# Antibiotic Treatment of Carbapenem-Resistant *Acinetobacter baumannii* Infections in View of the Newly Developed β-Lactams: A Narrative Review of the Existing Evidence

**DOI:** 10.3390/antibiotics13060506

**Published:** 2024-05-29

**Authors:** Francesca Serapide, Maurizio Guastalegname, Sara Palma Gullì, Rosaria Lionello, Andrea Bruni, Eugenio Garofalo, Federico Longhini, Enrico Maria Trecarichi, Alessandro Russo

**Affiliations:** 1Infectious and Tropical Disease Unit, Department of Medical and Surgical Sciences, ‘Magna Graecia’ University of Catanzaro, 88100 Catanzaro, Italy; francescaserapide@gmail.com (F.S.); sarapalma08@gmail.com (S.P.G.); rosarialionello0@gmail.com (R.L.);; 2Intensive Care Unit, Department of Medical and Surgical Sciences, ‘Magna Graecia’ University of Catanzaro, 88100 Catanzaro, Italy; andreabruni@unicz.it (A.B.); eugenio.garofalo@unicz.it (E.G.); flonghini@unicz.it (F.L.)

**Keywords:** *Acinetobacter baumannii*, carbapenem resistance, sulbactam/durlobactam, cefiderocol, guidelines, real-world evidence

## Abstract

It is estimated that antimicrobial resistance (AMR) is responsible for nearly 5 million human deaths worldwide each year and will reach 10 million by 2050. Carbapenem-resistant *Acinetobacter baumannii* (CRAB) infections represent the fourth-leading cause of death attributable to antimicrobial resistance globally, but a standardized therapy is still lacking. Among the antibiotics under consideration, Sulbactam/durlobactam seems to be the best candidate to replace current back-bone agents. Cefiderocol could play a pivotal role within combination therapy regimens. Due to toxicity and the pharmacokinetics/pharmacodynamics (PK/PD) limitations, colistin (or polymyxin B) should be used as an alternative agent (when no other options are available). Tigecycline (or minocycline) and fosfomycin could represent suitable partners for both NBLs. Randomized clinical trials (RCTs) are needed to better evaluate the role of NBLs in CRAB infection treatment and to compare the efficacy of tigecycline and fosfomycin as partner antibiotics. Synergism should be tested between NBLs and “old” drugs (rifampicin and trimethoprim/sulfamethoxazole). Huge efforts should be made to accelerate pre-clinical and clinical studies on safer polymyxin candidates with improved lung activity, as well as on the iv rifabutin formulation. In this narrative review, we focused the antibiotic treatment of CRAB infections in view of newly developed β-lactam agents (NBLs).

## 1. Introduction

The species belonging to *Acinetobacter baumannii*-*calcoaceticus* complex (ABC) are glucose-non-fermentative, aerobic Gram-negative coccobacilli [1]. The predominant predispositions to ABC infection are exposure to broad-spectrum antibiotics and disruption of anatomical barriers [2]. Among ABC, *A. baumannii* is the most common cause of human infections, responsible for a range of nosocomial infection across multiple anatomical sites, mainly ventilator-associated pneumonia (VAP) and central line-associated bloodstream infections (BSI) [3]. Peculiar findings of *A. baumannii* are the capacity to survive in unfavorable conditions, due to molecular features that promote environmental persistence (i.e., desiccation resistance, biofilm formation, and motility) and the ability to acquire or upregulate various resistance determinants (i.e., intrinsic and acquired β-lactamases, upregulation of efflux pumps, decreased outer membrane permeability, antibiotic target site modifications) [4]. Indeed, it is one of the microorganisms in the ESKAPE group, which have been identified as pathogens particularly characterized by increasing multiresistance and virulence dependent on mechanisms capable of evading the bactericidal action of antibiotics and is included in the list of critical priorities for antibiotic resistance among pathogens, along with *P. aeruginosa* (carbapenemase-resistant), *K. pneumoniae* and *Enterobacter* spp. (ESBL).

In addition, the World Health Organization (WHO) has recently included ESKAPE pathogens in the list of 12 bacteria for which new antibiotics are needed. Carbapenem resistance against *A. baumannii* varies between 30% and 80% and is commonly associated with the horizontal transfer of genes encoding oxacillinase (OXA) carbapenemases enzymes (including OXA-23 and OXA-24/40 enzymes) [5].

Carbapenem-resistant (CR)—*A. baumannii* infections are the fourth-leading cause of death attributable to antimicrobial resistance globally [6], but a standardized approach to antibiotic therapy is still lacking. Current CRAB treatment guidelines agree on the use of a combination regimen for severe infections, although there are some differences in the choice of antibiotic agents [7,8] (see Table 1).

The recent approval of sulbactam/durlobactam for the treatment of ABC pneumonia [9], together with the encouraging data from the real-world clinical use of cefiderocol [10,11], opens a new scenario for CRAB infections treatment.

In this narrative review, we focused the antibiotic treatment of CRAB infections in view of newly developed β-lactam agents (NBLs), presenting the available data, discussing the main hot points, and highlighting clinical questions awaiting further investigation.

## 2. Considerations and Available Data about New β-Lactam Agents

Beta-lactamases are classified into four groups (A, B, C, D) according to Ambler’s classification and into four categories (based on their biochemical function) according to Jacoby’s classification.

### 2.1. Sulbactam/Durlobactam

Sulbactam/durlobactam was recently approved in the U.S. for the treatment of pneumonia due to susceptible ABC [9]. Sulbactam is a class A β-lactamase inhibitor with intrinsic whole-cell activity against few bacterial species; its activity against *A. baumannii* is mediated through inhibition of the penicillin-binding protein (PBP)1a, PBP1b, and PBP3 [12]. However, sulbactam is susceptible to cleavage by several β-lactamases, such as TEM-1, Acinetobacter-derived cephalosporinase (ADC)-30, and OXAs (OXA-23, OXA-24/72, and OXA-58 families), hence its clinical utility for *A. baumannii* infections has been compromised over time [13]. Durlobactam is a novel non–β-lactam diazabicyclooctane β-lactamase inhibitor, with a broad-spectrum activity against class A, C, and D β-lactamases and PBPs, resulting in intrinsic antibacterial activity against *Enterobacterales* and restoration of β-lactam activity in (Multidrug Resistance) MDR Gram-negative pathogens [14]. In contrast to the other diazabicyclooctanes, durlobactam plays a crucial role in CRAB infection, due to the potent inhibition not only of class A β-lactamases but also of ADCs and OXAs, and the consequent ability to restore sulbactam activity. Furthermore, through PBP2 inhibition, it also showed a minimal intrinsic activity against the pathogen [14]. CRAB resistance to sulbactam/durlobactam was estimated to be 3.4%, mainly due to the substitution in the PBP3 determinant or the presence of New Delhi metallo-β-lactamase-1 (NDM-1); metallo β-lactamase (MBL)-producing strains are characterized by higher MIC values compared to other resistance mechanisms [15]. Although the current frequency of MBL-producing CRAB is relatively low [4,15], the commercialization of sulbactam/durlobactam may lead to an increase in the incidence of such strains, particularly NDM, for which *A. baumannii* has been considered a reservoir [2].

Sulbactam/durlobactam demonstrated a good intrapulmonary penetration ratio for epithelial lining fluid (ELF) to total plasma concentrations in healthy subjects, supporting the use of the combination in the treatment of pulmonary infections [16]. The ATTACK trial compared sulbactam/durlobactam (1 g/1 g over 3 h infusion q6h for 7–14 days) with colistin (2.5 mg/kg over 30 min infusion q12h for 7–14 days) for severe CRAB infection (mainly pneumonia, including VAP); both regimens were co-administered with imipenem/cilastatin as background therapy: sulbactam/durlobactam resulted non-inferior to colistin in 28-day all-cause mortality 19% (12/63) vs. 32% (20/62) presenting less adverse events (mainly nephrotoxicity, headache, nausea, and injection-site phlebitis) with respect to the competitor [17].

Co-administration of imipenem/cilastatin to provide coverage for eventual polymicrobial infections has raised questions about a possible synergic action of imipenem with sulbactam/durlobactam on CRAB, casting doubts on the real efficacy of the new β-lactam agent as monotherapy [18]. In vitro data on CRAB isolates of the ATTACK trial seem to indicate that imipenem does not provide a meaningful contribution to sulbactam–durlobactam activity against sulbactam–durlobactam-susceptible isolates [19]; however, a study conducted on 109 CRAB isolates in Greece showed that the addition of imipenem further lowered the MIC90 of sulbactam/durlobactam by one two-fold dilution [20]. Currently there are not ongoing trials on sulbactam/durlobactam [21].

The encouraging data seem to place sulbactam/durlobactam as a crucial player in CRAB infection, although two relevant clinical questions are still outstanding:(1)What is the best antibiotic partner to use in combination with sulbactam/durlobactam in CRAB infections?(2)Is sulbactam/durlobactam best used in combination treatment rather than monotherapy in CRAB infections?

In our opinion, studies addressing the first question should be prioritized for the following reasons: (i) the ATTACK trial tested sulbactam/durlobactam in a combination regimen, and the real role of imipenem addition has not yet been clearly elucidated; (ii) although there is no clear evidence of the superiority of a combination treatment over monotherapy in CRAB infection, currently the main guidelines agree on the use of combination treatment (at least for severe infections) [7,8]; (iii) the delay in placing in the therapy of cefiderocol in CRAB infections has been essentially due to unsatisfactory clinical results in trials testing the drug as monotherapy [22,23].

It would therefore be more prudent and appropriate, in our opinion, to prioritize the search for the best antibiotic partner of sulbactam/durlobactam and, subsequently, compare monotherapy vs. combination treatment.

### 2.2. Cefiderocol

Cefiderocol is a novel catechol-substituted siderophore cephalosporin, commercialized for the treatment of infections caused by CR Gram-negative bacteria (CR-GNB). Its activity is expressed through the inhibition of PBPs (primarily PBP3) and is transported into the periplasmic space mainly through the bacterial siderophore iron uptake system [24]. The unique structure of cefiderocol, along with low catalytic efficiencies of carbapenemases against this drug, confers stability against all four Ambler classes of β-lactamases [25]. The cefiderocol susceptibility rate for CRAB was estimated to be between 77.9% and 97.2% across different countries [25]. In the SIDERO surveillance program, 3.9% (204 out of 5225) of CRAB isolates had high (≥8 µg/mL) cefiderocol MICs without prior exposure to this antibiotic [26]: the β-lactamases Pseudomonas-extended resistance (PER) enzyme was detected in most of these isolates [25]. Although PER (and, to a lesser extent, NDM) contributes to increasing the cefiderocol MIC of *A. baumannii*, it does not alone confer resistance to the drug; cefiderocol resistance seems, in fact, to be mediated by the concomitant presence of other factors (such as PBP-3 modification, reduced expression of the siderophore receptor, and efflux overexpression) [27].

It is important to consider the phenomenon of heteroresistance, which is defined as a condition in which certain subpopulations of a biological sample exhibit varying degrees of phenotypic resistance, making in vitro identification of resistance difficult. In fact, some diagnostic methods may not be sensitive enough to identify these strains and may misclassify them as susceptible. However, by using appropriate antimicrobial susceptibility tests with higher inoculum, it is possible to detect even these subpopulations with intermediate or resistant minimal inhibitory concentrations.

In a study in the United States of 108 CRAB isolates, the frequency of resistant subpopulations was ≥1 in 106. In contrast, the resistance rate was 8% and the heteroresistance rate was 59% (64 isolates). In addition, the frequency of resistant subpopulations increased after exposure to cefiderocol and decreased after discontinuation of the drug. [28]. The authors of the survey hypothesized that heteroresistance could explain the discrepancy between the excellent in vitro susceptibility profile and the suboptimal clinical outcomes when used as monotherapy against CRAB [28]. However, no clinical data support this hypothesis; therefore, the real clinical impact of heteroresistance is yet unclear [29].

Cefiderocol showed an effective lung penetration in healthy subjects as well as in patients with pneumonia requiring mechanical ventilation [27]. The probability of target attainment (PTA) for 100% fT > MIC was >90% across all infection sites for pathogens with cefiderocol MICs of ≤4 µg/mL; adequate plasma exposure can be achieved at the drug recommended dosing regimen (2 g over 3 h infusion q8h) for the infected patients [30].

Despite the in vitro efficacy and PK/PD characteristics, clinical results on cefiderocol monotherapy were unsatisfactory [22,23]. The CREDIBLE-CR trial compared cefiderocol with the best available treatment (BAT) for CR-GNB: in the subgroups of patients with CRAB, all-cause mortality at day 28 (± 3 days) after the end of treatment was 49% (19/39) in patients treated with cefiderocol vs. 18% (3/17) in those treated with BAT; however, baseline risk for mortality was higher in the cefiderocol group and BAT was largely heterogenous [22]. The APEKS-NP trial compared cefiderocol with meropenem (high-dose, extended-infusion) for the treatment of Gram-negative nosocomial pneumonia: all-cause mortality in patients with CRAB infection at day 14 were 12.4% (18/145) vs. 11.6% (17/146) in the cefiderocol and meropenem groups, respectively [23].

A prospective, observational, single-centre study compared the clinical failure (defined as the need to switch to second-line antibiotic therapy due to lack of clinical response or recurrence of VAP up to 7 days from the end of active therapy) between cefiderocol and colistin-based regimen groups, in non-COVID-19 ICU patients with CRAB [31]. Cefiderocol was administered as a combination treatment in 52.5% (21/40) of the cases, and as monotherapy in the remaining 19 cases. Multivariable Cox regression analysis showed that a first-line cefiderocol-based regimen was an independent protective factor in clinical failure risk (HR 0.38, 95% CI 0.18–0.76, *p* = 0.007), the result confirmed through the IPTW analysis. However, there was no significant difference in 28-day all-cause mortality between groups: 35% (14/40) vs. 52% (26/50) in cefiderocol and colistin-regimen groups, respectively [31]. Four Italian retrospective studies compared cefiderocol with colistin-based regimen in ICU patients with CRAB [10,11,32,33] (see Table 2)

Two of the studies demonstrated a comparable rate of all-cause mortality between treatment groups at days 28 and 30, respectively [32,33]. In contrast, the cefiderocol-based regimen was identified as an independent predictor of 30-day survival in the remaining studies [10,11].

In a recent study aimed at evaluating the clinical efficacy of cefiderocol-based regimens for the treatment of CRAB infections, a meta-analysis was conducted on the available data (five abovementioned observational studies and the CREDIBLE-CR trial): a trend was demonstrated towards a significantly lower mortality rate in patients who received a cefiderocol-based compared to a colistin-based regimen [34]. Interestingly, after excluding studies at high/severe risk of bias and considering only studies performing proper adjustments for confounders, cefiderocol-based regimens were associated with a significantly lower risk of mortality (N = 4; OR 0.53; 95% CI 0.39–0.71; I2 = 0.0%) [34]; however, the number of included studies was limited. Unfortunately, no subgroup analysis according to cefiderocol mono- or combination therapy was performed due to a lack of available adjusted data [34]. Anyhow, it is interesting to note that among the mentioned four retrospective studies [10,11,29,30], cefiderocol was an independent predictor of 30-day survival in the studies where the rate of cefiderocol combination-regimen was higher (see Table 2). Furthermore, in the study by Falcone et al., the microbiological failure rate was significantly higher in patients who received monotherapy compared to those who received combination therapy 42.9% (6/14) vs. 6.3% (2/32) and four patients with microbiological failure developed resistance to cefiderocol [10].

The conflicting data between the CREDIBLE-CR trial and observational studies could be explained, and the net of the biases already described, by the fact that in real-world studies, cefiderocol was administered more frequently in a combination treatment. A lower efficacy in monotherapy could in turn be explained by the phenomenon of heteroresistance. Therefore, the real clinical impact of cefiderocol heteroresistance needs to be investigated.

While available data discourage the use of cefiderocol monotherapy in CRAB infections, real-world data on its use as a back-bone agent in combination treatment are encouraging. To accelerate the right placement in therapy of such a compound, priority should be given to studies that address the following questions:(1)Is there a role for cefiderocol as a back-bone agent in combination treatment for CRAB infections?(2)What is the best antibiotic partner to use in combination with cefiderocol in CRAB infections?

A trial comparing cefiderocol + ampicillin-sulbactam vs. colistin ± meropenem for CRAB infections is currently registered, although not yet recruiting (NCT05922124) [21].

Of interest, a case of VAP due to extremely-drug-resistant (XDR)-*A. baumannii* successfully treated with cediferocol + sulbactam/durlobactam has been reported [35]. Probably, rather than as a combination partner for sulbactam/durlobactam, cefiderocol might be better used as an alternative back-bone regimen when sulbactam/durlobactam is contraindicated (i.e., resistance or intolerance). As a matter of fact, cefiderocol is the only known option against MBL-CRAB. A cefiderocol-sparing approach could prove to be a winning strategy in case of an increase in the rate of CRAB resistance to sulbactam/durlobactam, an event that is likely to occur after the drug’s worldwide commercialization.

Randomized clinical trials (RCTs) to evaluate the efficacy of these two NBLs in combination treatment for CRAB infections are urgently needed.

## 3. Place in Therapy of Traditional Agents for Treatment of CRAB

### 3.1. Polymyxins

This antibiotic acts by binding to the anionic molecules of LPS, displacing Mg^2+^ and Ca^2+^ from the outer cell membrane of Gram-negative bacteria, causing permeability changes in the cell envelope and leakage of cell contents. The mechanism of resistance in *A. baumannii* is determined by the complete loss of lipopolysaccharide production.

Colistin, available intravenously as the prodrug colistimethate sodium, and polymyxin B have been the most used therapeutic options for CRAB [36]. Despite colistin resistance being uncommon in *A. baumannii*, there has recently been a worldwide increase in the resistance rate, reaching a peak of approximately 10% in Europe [37].

Polymyxins lung penetration is suboptimal; furthermore, colistin has several disadvantages compared to polymyxin B in terms of pharmacokinetic characteristic: due to the prodrug administration, colistin plasma concentration rises slowly, is subjected to a greater inter-patient variability and, in patients with normal renal function, the target plasma concentration is difficult to achieve [38]. Therefore, except for urinary tract infections (UTIs), polymyxin B is preferred to colistin for severe infections [38]; however, its availability is limited globally. On the contrary, colistin is preferred in UTIs due to its higher urinary concentration (polymyxin B is extensively reabsorbed by the renal tubular cells) [39]; moreover, it seems to present fewer side effect than polymyxin B when administered by inhalation [40]. However, data on the real clinical utility of polymyxins inhalation in patients with CRAB pneumonia are conflicting [40], as are the recommendations of professional societies [8,9,38]. The main adverse effect of polymyxins administration is nephrotoxicity, which in some reports reaches the rate of 55%, while neurotoxicity is less common [40].

Although the combination of colistin with other antibiotics (rifampicin, fosfomycin, meropenem), produced an in vitro synergistic effect against *A. baumannii*, clinical trials failed to demonstrate improved efficacy of colistin in combination with these antibiotics compared to monotherapy [41,42,43]. Several safer polymyxin candidates, with improved activity (also in term of lung penetration) compared to colistin and polymyxin B, are undergoing preclinical and clinical evaluations [39]. Until their commercialization, it would be appropriate, in our opinion, to use polymyxins as alternative anti-CRAB agents (when no other options are available). In conclusion, the drug in question has been found to have certain limitations in terms of its use. These include nephrotoxicity [40], suboptimal pulmonary penetration, and suboptimal plasma concentrations [38]. It is regarded as a potential alternative agent in instances where no other viable options exist.

### 3.2. Tetracycline Derivatives

The pharmacological action of these drugs is exerted through the inhibition of the 30S ribosomal subunit, thereby impeding protein synthesis. *A. baumannii* resistance is mediated by three distinct mechanisms: (i) ATP-dependent efflux, (ii) inactivation of tetracyclines by enzymes, and (iii) ribosomal protection proteins (RPPs).

Tigecycline and minocycline, capable of escaping common tetracycline resistance mechanisms, are currently recommended for CRAB infections treatment [7,8]. Clinical breakpoints for tigecycline against *Acinetobacter* spp. have not been established [8], while the susceptibility rate to minocycline is about 85%, dropping to around 70% for multi-drug resistant (MDR) isolates [44].

Although minocycline seems to have better lung penetration than tigecycline [45,46], both compounds are characterized by suboptimal exposures in blood and serum [5]. Therefore, high-dose administration is recommended for CRAB infections [7,8]. However, in a PTA analysis, the high-dose minocycline regimen currently employed in clinical practice was predicted to result in a suboptimal plasma *ƒ*AUC:MIC profile for patients with *A. baumannii* infections with MICs > 1 mg/L; among MDR-ABC, minocycline MIC values were >1 mg/L for 60% of the tested isolates [45]. Such data would call into question the current susceptibility breakpoint for minocycline (≤4 mg/L), based on the rat pneumonia model [45]. Furthermore, the global availability of intravenous minocycline formulation is limited. Regarding the PK/PD data on high-dose tigecycline, a study on serum and ELF concentrations among critically ill patients was conducted: PK/PD target attainment for pneumonia was ≥75% with MICs ≤ 0.5 mcg/mL [46], but only 31% of international CRAB isolates demonstrated tigecycline MICs ≤ 0.5 [33].

However, in vitro synergism between tigecycline and cefiderocol has been demonstrated among both cefiderocol-resistant and susceptible CRAB isolates [47]; moreover, in observational studies, tigecycline was one of the most frequently administered agents in a cefiderocol-combination regimen [34]. Tigecycline (or minocycline where iv formulation is available) could represent a suitable partner for both NBLs in CRAB infections treatment. RCTs are needed to confirm this hypothesis.

Preclinical and clinical data on the novel tetracycline derivatives, eravacycline and omadacycline, suggest reduced activity against CRAB compared to tigecycline and minocycline [8]; however, data are scarce and further studies are necessary to understand if and what role they may have in CRAB infections. It is important to note that suboptimal efficacy has been observed in serum, lung, and urine samples [5].

### 3.3. Fosfomycin

Fosfomycin exerts its antimicrobial effect by inhibiting the synthesis of peptidoglycan, a component of the bacterial cell wall, at an earlier stage than betalactams. *A. baumannii* develops resistance to fosfomycin through different mechanisms, including the presence of the fosfomycin efflux transporter MFS-encoded AbaF and fosfomycin resistance glutathione transferase.

Fosfomycin is characterized by good tissue penetration (including infected lung tissue) as well as a good concentration in serum, urine, and cerebrospinal fluid, with a good safety profile [48]. ABC is intrinsically resistant to fosfomycin; however, it resulted in a synergistic effect in vitro when combined with several antibiotics against CRAB [48,49].

Due to a single trial which failed to demonstrate the superiority of colistin plus fosfomycin over colistin alone [43], but also because intravenous fosfomycin is not available in the U.S., its role in CRAB treatment has been under-investigated [49] and the drug is not currently recommended for such infections [7,8]. Regardless, in a prospective, observational, multicentre study, conducted on 180 patients with HAP due to MDR—*A. baumannii*, 44 patients were treated with a fosfomycin-containing regimen (29 in double and 15 in triple combination regimen) which turned out to be a factor associated with 30-day survival (*p* < 0.001) [50]. Moreover, in a case-series study conducted on 20 ICU patients with BSI due to pan-drug resistant (PDR)—*A. baumannii*, a fosfomycin-containing regimen (one case in double regimen and seven cases in regimens including at least three antibiotics) was associated with 28-day survival (*p* < 0.005) [51].

Regarding the potential role as a partner for NBLs, in vitro data showed a synergistic effect of fosfomycin in combination with cefiderocol against a cefiderocol-resistant CRAB isolate [52]. Furthermore, in observational studies, fosfomycin was the most frequently administered agent in a cefiderocol-combination regimen [34].

Interestingly, in one of the abovementioned retrospective studies, fosfomycin was co-administered in 14 out 19 patients of the cefiderocol group, resulting an independent factor of 30-day survival (*p* < 0.001); however, 8 out of 14 patients received fosfomycin in a combination of at least three antibiotics [11].

Finally, a study conducted on sulbactam-resistant CRAB isolates in a hollow-fiber infection model showed that a combination of fosfomycin and extended infusion of sulbactam produced a 4 log_10_ reduction in colony count within 24 h, followed by suppression of regrowth [53]. Although fosfomycin is not currently recommended, it could represent a suitable partner for both NBLs in CRAB infections treatment. RCTs are needed to confirm this hypothesis.

### 3.4. High-Dose Extended-Infusion Meropenem

A high-dose extended-infusion meropenem regimen enhances the PK exposure of the compound, but considering the high meropenem MIC values of CRAB isolates, it does not reach the optimal cumulative fraction of response in such infections [54]. However, a triple combination regimen containing meropenem has been used successfully for XDR-CRAB infections [5] and, in one case, meropenem was co-administered with cefiderocol [55].

Interestingly, cefiderocol + meropenem demonstrated an in vitro synergistic effect against 79.5% of cefiderocol-resistant isolates [47]. A case of PDR-CRAB infection was cured with a combination of sulbactam/durlobactam + meropenem; of note, the addition of meropenem reduced the sulbactam/durlobactam MIC from 8 to 4 mg/L (the preliminary susceptibility breakpoint) [56]. If further studies confirm the synergism between meropenem and NBLs, high-dose extended-infusion meropenem + sulbactam/durlobactam (or cefiderocol) could represent a valuable option for PDR-CRAB infections.

From this perspective, a meropenem-sparing strategy should be adopted in case of CRAB infections due to strains sensitive to the NBLs. In case of co-administration of high-dose meropenem with another β-lactam, a close monitoring of side effects should be warranted, due to the high probability of an increase in side effects (i.e., epilepsy).

### 3.5. Aminoglycosides

These antibiotics act by binding to the 30S subunit of ribosomes, thereby inhibiting protein synthesis in bacteria. The resistance mechanism results in enzymatic modification of the aminoglycoside molecule, primarily through *N*-acetylation, *O*-nucleotidylation, or *O*-phosphorylation at various points along the molecule.

Aminoglycosides are currently recommended in combination treatment for susceptible CRAB isolates [7]; however, the global resistance rate is >80% [37]. Furthermore, considering drugs’ side effects [57] and suboptimal ELF concentrations [58], their prospective role in CRAB infections will likely be limited to a few selected cases.

### 3.6. Rifamycins

Rifamycins act as inhibitors of bacterial DNA-dependent RNA polymerase. *A. baumannii* resistance is the result of mutations in the rpoB gene, which encodes the beta-subunit of rifamycin-sensitive RNA polymerase. These mutations prevent RNA elongation immediately after the addition of the first nucleotides.

Rifampicin showed synergistic effect against MDR-*A. baumannii* when combined with colistin [59], but the clinical efficacy of such treatment has not been demonstrated [42] and rifampicin is currently not recommended in CRAB infections [7,8]. However, in vitro synergism seems to depend by rifampicin MICs, and data on rifampicin MIC values of CRAB isolates are scant [59].

Interestingly, in vitro synergism between rifampicin and sulbactam against CRAB isolates has been demonstrated [60]; moreover, in a case series on 12 infant and young children with severe VAP caused by XDR-*A. baumannii*, the combination of rifampicin and sulbactam appeared to be an effective and safe therapy (9 out of 12 patients were considered cured) [61]. Studies addressing an in vitro synergic effect between rifampicin and the NBLs should be conducted, eventually followed by clinical evaluations of combined treatments.

Rifabutin, available only as an oral formulation, overcomes the common rifamycin resistance mechanisms of *A. baumannii*, displaying potent in vitro activity against CRAB strains [62]. The drug is rapidly distributed in all organs and tissues, where levels are constantly higher than plasma levels; unfortunately, the oral bioavailability of rifabutin is very low, limiting its clinical utility [62]. An intravenous formulation of rifabutin (BV100) has been developed to maximize clinical efficacy against *A. baumannii* infections while minimizing the risk of resistance development, and it is currently under investigation in a Phase I clinical trial [62].

### 3.7. Trimethoprim/Sulfamethoxazole

Trimethoprim is a dihydrofolate reductase (DHFR) inhibitor, which blocks the formation of tetrahydrofolic acid by dihydrofolic acid. Sulfonamides, on the other hand, are known to inhibit dihydropteroate synthase (DHPS). The presence of trimethoprim-resistant dihydrofolate reductase A. baumannii is considered resistant.

Sporadic cases of MDR—*A. baumannii* have been successfully treated with a combination regimen including trimethoprim/sulfamethoxazole [11,63], while in a retrospective match cohort study, comparing trimethoprim/sulfamethoxazole monotherapy with other regimens for CRAB infections, all-cause 30-day mortality was lower in trimethoprim/sulfamethoxazole group (24.5%, 13 of 53 vs. 38.6%, 32 of 83); however, the baseline risk for mortality was higher in the comparator antibiotics group [64]. The resistance rate among CRAB isolates is estimated to be >80% [63] and trimethoprim/sulfamethoxazole is currently not recommended for such infections [8,9].

Possible synergistic effects between trimethoprim/sulfamethoxazole and the NBLs should be investigated; the compound could represent a valuable partner in CRAB combination-treatment, at least for infections due to sensitive isolates.

### 3.8. Novel Antibiotics (Zosurazalpin)

Among the most interesting and promising options not yet on the market, zosurazalpin has been identified as a drug with potential antibacterial activity against CRAB in vitro and in mouse models, capable of overcoming resistance mechanisms. The mechanism of action involves inhibition of the LptB2FGC complex by blocking the transport of bacterial lipopolysaccharide from the inner membrane to its target on the outer membrane.

The main characteristics of antibiotics are summarized in Table 3.

## 4. Conclusions

The commercialization of NBLs opens a new scenario for the treatment of CRAB infections.

However, data are limited, and relevant clinical questions remain outstanding, such as the superiority of a combination treatment over monotherapy and the best antibiotic partner to use in combination.

In this narrative review, we have highlighted the main characteristics and potential roles of anti-CRAB antibiotics (summarized in Table 3), in view of recent data on NBLs. Sulbactam/durlobactam seems to be the best candidate to replace current back-bone agents. Cefiderocol, despite the suboptimal efficacy in monotherapy, could play a crucial role in combination-regimen (probably as an alternative back-bone agent): RCTs are needed to better evaluate the role of these new β-lactams in CRAB infections. Due to toxicity and the PK/PD limitations, colistin should be used as an alternative anti-CRAB agent (when no other options are available).

Tigecycline and fosfomycin could represent suitable partners for both NBLs: RCTs –comparing the efficacy of these two drugs as partner-antibiotics in combined CRAB treatment should be prioritized [65]. High-dose extended-infusion meropenem, if further studies will confirm the synergistic effect between meropenem and NBLs’ should be taken into consideration in case of PDR-CRAB infections, while the role of aminoglycosides will be probably limited to a few selected cases. In vitro data on the synergism between NBLs and rifampicin or trimethoprim/sulfamethoxazole are needed to guide eventual any future clinical investigations into the possible role of these “old antibiotics” as partner-agents in CRAB infection treatment.

Taking in account the limitations of current available partner-antibiotic candidates for CRAB infections, huge efforts should be made to accelerate pre-clinical and clinical studies on safer polymyxin candidates with improved lung activity, as well as on the intravenous formulation of rifabutin or about new promising molecules like cefoperazone-sulbactam [66]. Important strategies about carbapenem-sparing could be another important point to reduce the spread of CRAB strains [67].

## Figures and Tables

**Table 1 antibiotics-13-00506-t001:** Differences between CRAB treatment guidelines.

	ESCMID Guidelines (April 2022)	IDSA Guidance (July 2023)
Combination antibiotic regimen	For severe and high-risk CRAB infection	For moderate–severe CRAB infection
Ampicillin/sulbactam	For patients with CRAB susceptible to sulbactam and HAP/VAP (1 g sulbactam component q6h)	Back-bone treatment for all CRAB infection (6–9 g sulbactam component daily)
Polymyxins	Either colistin or polymyxin B: for patients with CRAB resistant to sulbactam susceptible to polymyxins; in combination with one other in vitro active agent for severe, susceptible to polymyxins, CRAB infection	Polymyxin B in combination with at least one other agent for the treatment of CRAB infections (Colistin only for CRAB UTIs)
Tetracycline derivatives	High-dose tigecycline: for patients with CRAB resistant to sulbactam susceptible to tigecycline; in combination with one other in vitro active agent for severe, susceptible to tigecycline, CRAB infection	High-dose minocycline (preferred option) or high-dose tigecycline in combination with at least one other agent for the treatment of CRAB infections
Cefiderocol	Not recommended	In combination with at least one other agent for the treatment of CRAB infections refractory to other antibiotics (or when the use of other antibiotics is precluded)
Aminoglycosides	In combination with one other in vitro active agent for severe, susceptible to aminoglycosides, CRAB infection	Not recommended
High-dose extended-infusion meropenem	In combination with one other in vitro active agent for severe CRAB infections with a meropenem MIC < 8 mg/L	Not recommended

Legend. CRAB: carbapenem-resistant *Acinetobacter baumannii*; ESCMID: European Society of Clinical Microbiology and Infectious Diseases; IDSA: Infectious Diseases Society of America; HAP: health-care associated pneumonia; VAP: ventilator-associated pneumonia; UTIs: urinary tract infections. High-dose tigecycline: 200 mg as a loading dose followed by 100 mg q12h; High-dose minocycline: 200 mg q12h. High-dose extended-infusion meropenem: 2 g over 3 h infusion q8h.

**Table 2 antibiotics-13-00506-t002:** Retrospective observational studies comparing cefiderocol with colistin-based regimen in ICU patients with CRAB.

	Pascale et al. [32] Multicentre (January 2020–April 2021)	Mazzitelli et al. [33] Single-Centre (August 2020–July 2022)	Falcone et al. [10] Single-Centre (January 2020–August 2021)	Russo et al. [11] Single-Centre (March 2020–August 2022)
Population: antibiotic-based regimen groups	107 patients: 42 CFD 65 COL	111 patients: 60 CFD 51 COL	124 patients: 47 CFD 77 COL	73 patients: 19 CFD 54 COL
COVID-19 coinfection	100%	32%	38.7%	100%
Site of infection	BSI (58%) LRTI (41%) Others (1%)	BSI (47.7%) Pneumonia (52.3%)	BSI (57.4%) VAP (25.5%) Others (17%)	VAP and concomitant BSI (100%)
Patients received CFD in combination	0	30 (50%)	33 (70%)	19 (100%)
Main agents co-administered with CFD	/	TGC (18/30) MEM (13/30) FOS (8/30)	TGC (21/33) FOS (8/33)	FOS (7/19) FOS + TGC (7/19) TGC (1/19)
28–30 day all-cause mortality: CFD group vs. COL group	23 (55%) vs. 38 (58%) (*p*-value: 0.7)	26 (51%) vs. 22 (37%) (*p*-value: 0.13)	16 (34%) vs. 43 (56%) (*p*-value: 0.018)	6 (31.5%) vs. 53 (98%) (*p*-value < 0.001)

Legend. ICU: intensive care unit; CRAB: carbapenem-resistant *Acinetobacter baumannii*; CFD: cefiderocol; COL: colistin; BSI: bloodstream infection; LRTI: low respiratory tract infection; VAP: ventilator-associated pneumonia; TGC: tigecycline; MEM: meropenem; FOS: fosfomycin.

**Table 3 antibiotics-13-00506-t003:** Potential role of anti-CRAB antibiotics.

	Potential Role	Main Mechanisms of Action	Main Mechanisms of Resistance	Evidences (or Available Data)	Limits	Studies to Be Prioritized
Sulbactam/ durlobactam	Back-bone agent in combination treatment	Inhibition of penicillin binding proteins 1 and 3 (involved in synthesis of bacterial peptidoglycan)/Uses a reversible mechanism of inhibition through β-lactamase active site carbamoylation	Single amino acid changes near the active site serine of PBP3 (S336), the target of sulbactam	RCT: non-inferior to COL (both co-administered with IPM-CLN) [17]	Efficacy as monotherapy not known	RCTs finding the best partner-agent
Cefiderocol	Back-bone agent in combination treatment	Utilizes the siderophore–iron complex pathway to penetrate the outer membrane of Gram-negative organisms in addition to normal passive diffusion through membrane porins	Alterations of the intrinsic AmpC and siderophore receptors	Metanalysis: lower risk of mortality rate compared to COL-based regimen [34]	Unsatisfactory efficacy as monotherapy when compared to COL [22] and MEM [23]	-RCTs confirming the role as back-bone agent; -RCTs finding the best partner-agent
Polymyxins	COL (or PB): alternative agent (when no other options are available)	Binds with the anionic LPS molecules by displacing Mg^2+^ and Ca^2+^ from the outer cell membrane of Gram-negative bacteria, leading to permeability changes in the cell envelope and leakage of cell contents	Complete loss of Lipopolysaccharide production	Large clinical experience as back-bone agent [36]. (Data on combination with NBLs are missing. )	-Nefrotoxicity [40]; -suboptimal lung penetration [38]; -suboptimal plasma concentrations [38].	Accelerate studies on safer polymyxin with lung improved activity
Tetracycline derivatives	High-dose TGC (or MNC): partner-agent in combination treatment	Inhibit the 30S ribosomal subunit and thereby inhibit protein synthesis	(i) Efflux dependent on ATP, (ii) inactivation of tetracyclines by enzymes, and (iii) ribosomal protection proteins (RPPs)	TGC + CFD: -in vitro synergism [47]. (one of the most frequently used combination in observational studies [34])	Suboptimal exposures in serum, lung and urine [5]	RCTs comparing TGC and FOS as partner-agent
Fosfomycin	Partner-agent in combination treatment	Inhibition of bacterial cell wall peptidoglycan synthesis at an earlier stage than betalactams.	(i) fosfomycin efflux MFS transporter AbaF encoded; (ii) fosfomycin resistance glutathione transferases	-Retrospective study: associated with 30-day survival in combination with CFD [11]; -In vitro synergism with CFD [52] and SUL [53]; (the most commonly used agent in combination with CFD in observational studies [34])	-Data coming from the observational study included regimens of more than 2 agents [11] -*A. baumannii* is intrinsically resistant to the drug [48]	RCTs comparing TGC and FOS as partner-agent
High-dose extended- infusion meropenem	Partner-agent in PDR-CRAB infections (to be spared in treatment of strains sensitives to NBLs)	Binds penicillin-binding protein (PBP) in the bacterial cell wall and inhibits peptidoglycan cross-linking associated with cell wall synthesis	Production of enzymes such as beta-lactamases	In vitro synergism against CFD-resistant strains [47]. (Combined with SUL-DUR: a single case report of PDR-CRAB cured, with in vitro synergistic effect [56])	-Suboptimal cumulative fraction of response [54]; -possible increase in side effects rate if co-administered with other BLs	-In vitro studies on synergism with NBLs; -Clinical studies on PDR-CRAB infections
Aminoglycosides	Alternative partner-agent for few selected cases	Bind to the 30S subunit of ribosomes, inhibiting protein synthesis in bacteria.	Enzymatic modification primarily through N-acetylation, O-nucleotidylation, or O-phosphorylation at different locations of the aminoglycoside molecule.	Currently recommended as a combination treatment for susceptible CRAB isolates [7]. (Data on combination with NBLs are missing)	-Resistance rate among CRAB isolates > 80%; -suboptimal concentration in lung [37]; -high rate side effects [57]	*/*
Rifamycins	Alternative partner-agent	Inhibit bacterial DNA-dependent RNA polymerase	Mutations in the rpoB gene, which encodes rifamycin sensitive beta-subunit of RNA polymerase and averts RNA elongation just after adding the first nucleotides	RFM + SUL: -in vitro synergism [60]; (a case series on 12 pediatric patients reported clinical efficacy in VAP due to XDR-*A. baumannii* [61])	Synergism seems to depend by rifampicin MICs, but MICs data are scant [59]	-Accelerate clinical studies on rifabutin iv formulation; -in vitro studies on synergism between RFM and NBLs
Trimethoprim/ sulfamethoxazole	Alternative partner-agent	Trimethoprim is a dihydrofolate reductase (DHFR) inhibitor (blocking tetrahydrofolic acid formation by dihydrofolic acid), while sulfonamides are known dihydropteroate synthase (DHPS) inhibitors	Trimethoprim-resistant dihydrofolate reductases	(Successfully administered in combination with CFD in sporadic cases [11])	Resistance rate among CRAB isolates > 80% [63]	In vitro studies on synergism with NBLs

Legend. CRAB: carbapenem-resistant *Acinetobacter baumannii*; RCT: randomized clinical trial; COL: colistin; IMP-CLN: imipenem/cilastatin; MEM: meropenem; PB: polymyxin B; NBLs: newly developed β-lactam agents; TGC: tigecycline; MNC: minocycline; CFD: cefiderocol; FOS: fosfomycin; SUL: sulbactam; *A. baumannii*: *Acinetobacter baumannii*; PDR: pan-drug resistant; DUR: durlobactam; BLs: β-lactam agents; RFM: rifampicin; VAP: ventilator-associated pneumonia; XDR: extremely drug resistant; MICs: minimal inhibitory concentrations. High-dose tigecycline: 200 mg as a loading dose followed by 100 mg q12h; high-dose minocycline: 200 mg q12h; high-dose extended-infusion meropenem: 2 g over 3 hrs. infusion q8h.

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
