# Peer review of "Antibiotic Treatment of Carbapenem-Resistant Acinetobacter baumannii Infections in View of the Newly Developed β-Lactams: A Narrative Review of the Existing Evidence"

_antibiotics, 2024, doi:10.3390/antibiotics13060506_

Round 1

Reviewer 1 Report

Comments and Suggestions for Authors

Dear Authors,

The manuscript present a complex, up-to-date review article on antibiotic resistance of the microorganism Acinetobacter baumannii.

In the Introduction section, I recommend the authors to provide some additional information regarding the particular pathogenicity of this bacterium, a gram-negative microorganism considered by the WHO to be the most dangerous superbacterium, responsible for numerous nosocomial infections, also classified in the category of micro-organisms in a critical stage aiming at the need for the development of active antibiotics, or to approach some strategies to combat this bacterium (along with Pseudomonas Aeruginosa and Enterobacteriaceae).

The first mention of an abbreviated term requires the exposure of the non-abbreviated term (PK/PD, MDR).

Recent documentation, appropriate citations, that support the data and studies presented in the article.

Line 143 - Mention of the phenomenon of heteroresistance requires a brief elaboration, considering that the concept of heteroresistance represents the heterogeneous susceptibility of a species of microorganisms (including other highly pathogenic bacterial species, such as Klebsiella, Enterobacteriaceae, etc.) to an antibiotic, some subpopulations may be resistant to the antibiotic, while others are sensitive.

Lines 330-331 - the mention of meropenem toxicity should be supported by a citation, considering that this antibiotic, unlike other carbapenems, has a lower toxicity (the structure, with the methyl substituent, makes it resistant to hydrolysis by renal dehydropeptidases 1).

The authors present several classes of valuable antibiotics through their action and potential use in the case of CRAB infections. I suggest that it be supplemented with two more new antibiotics, with important action, at least until this moment in CRAB infections, such as:

- Zosurabalpin, an antibiotic developed in collaboration between the pharmaceutical company Roche and Harvard University, for the treatment of Acinetobacter baumannii (CRAB) - carbapenem-resistant Acinetobacter baumannii, with a mechanism involving a lipopolysaccharide transporter. Structurally, this antibiotic is a macrocyclic peptide with strong antibacterial activity on carbapenem-resistant Acinetobacter baumannii – it blocks the transport of bacterial lipopolysaccharides from the inner membrane to its destination on the outer membrane by inhibiting the LptB 2 FGC complex.

- Iboxamycin – effective antibiotic against ESKAPE bacteria (including Acinetobacter baumannii), which from a structural point of view belongs to the macrobicyclic oxepano-proline-amides class, developed starting from the antibiotic clindamycin, with a similar structure to lincosamides.

Author Response

Dear Authors,

The manuscript present a complex, up-to-date review article on antibiotic resistance of the microorganism Acinetobacter baumannii.

R: Dear reviewer, Thank you for your valuable suggestions and corrections, which have enabled us to further enrich the manuscript and make it more usable and accessible to the reader. We have made the changes requested, as detailed below, to make the content more comprehensive.

In the Introduction section, I recommend the authors to provide some additional information regarding the particular pathogenicity of this bacterium, a gram-negative microorganism considered by the WHO to be the most dangerous superbacterium, responsible for numerous nosocomial infections, also classified in the category of micro-organisms in a critical stage aiming at the need for the development of active antibiotics, or to approach some strategies to combat this bacterium (along with Pseudomonas Aeruginosa and Enterobacteriaceae).

The first mention of an abbreviated term requires the exposure of the non-abbreviated term (PK/PD, MDR).

Recent documentation, appropriate citations, that support the data and studies presented in the article.

Line 143 - Mention of the phenomenon of heteroresistance requires a brief elaboration, considering that the concept of heteroresistance represents the heterogeneous susceptibility of a species of microorganisms (including other highly pathogenic bacterial species, such as Klebsiella, Enterobacteriaceae, etc.) to an antibiotic, some subpopulations may be resistant to the antibiotic, while others are sensitive.

Lines 330-331 - the mention of meropenem toxicity should be supported by a citation, considering that this antibiotic, unlike other carbapenems, has a lower toxicity (the structure, with the methyl substituent, makes it resistant to hydrolysis by renal dehydropeptidases 1).

The authors present several classes of valuable antibiotics through their action and potential use in the case of CRAB infections. I suggest that it be supplemented with two more new antibiotics, with important action, at least until this moment in CRAB infections, such as:

- Zosurabalpin, an antibiotic developed in collaboration between the pharmaceutical company Roche and Harvard University, for the treatment of Acinetobacter baumannii (CRAB) - carbapenem-resistant Acinetobacter baumannii, with a mechanism involving a lipopolysaccharide transporter. Structurally, this antibiotic is a macrocyclic peptide with strong antibacterial activity on carbapenem-resistant Acinetobacter baumannii – it blocks the transport of bacterial lipopolysaccharides from the inner membrane to its destination on the outer membrane by inhibiting the LptB 2 FGC complex.

- Iboxamycin – effective antibiotic against ESKAPE bacteria (including Acinetobacter baumannii), which from a structural point of view belongs to the macrobicyclic oxepano-proline-amides class, developed starting from the antibiotic clindamycin, with a similar structure to lincosamides.

R: -      Lines 62-67, page 2: We have provided, as you suggested, a more complete description of the pathogenicity of the bacterium.

-           The unabbreviated term was added before the abbreviations.

-           Lines 186-192, page 5: The concept of heteroresistance has been expanded, as requested

-           Line 428, page 10: Meropenem appears to be one of the safest drugs, even at high doses, but in critically ill, frail patients with multiple comorbidities, who are inherently more susceptible to the development of common and uncommon adverse effects (also listed in the Fact Sheet), combination treatment with meropenem and high-dose beta-lactams could be correlated with worse survival. Therefore, the goal is to optimize the blood concentrations of the drugs as much as possible to minimize the risk of poor prognosis. However, for a better understanding of the concept, we will modify the term "toxicity" to "side effects".

-           Lines 496-502, page 12: We proceeded to include the proposed new treatment option.

Reviewer 2 Report

Comments and Suggestions for Authors

To consider adding sulperazone and doxycycline review 

Brazilian Journal of Microbiology (2023) 54:1795–1802

https://doi.org/10.1007/s42770-023-01015-0

Oral doxycycline to carbapenem‑resistant Acinetobacter baumannii infection as a polymyxin‑sparing strategy: results from a retrospective cohort

Felipe Francisco Tuon 1  · Carolina Hikari Yamada 1  · Ana Paula de Andrade 1  · Lavinia Nery Villa Stangler Arend 1  · Dayana dos Santos Oliveira 1  · João Paulo Telles 2,3 

Antimicrobial Activity of Cefoperazone-Sulbactam Tested against Gram-Negative Organisms from Europe, Asia-Pacific, and Latin America

Helio S. Sader

PlumX Metrics

DOI: https://doi.org/10.1016/j.ijid.2019.11.006

Author Response

To consider adding sulperazone and doxycycline review 

Brazilian Journal of Microbiology (2023) 54:1795–1802

https://doi.org/10.1007/s42770-023-01015-0

Oral doxycycline to carbapenem‑resistant Acinetobacter baumannii infection as a polymyxin‑sparing strategy: results from a retrospective cohort

Felipe Francisco Tuon 1  · Carolina Hikari Yamada 1  · Ana Paula de Andrade 1  · Lavinia Nery Villa Stangler Arend 1  · Dayana dos Santos Oliveira 1  · João Paulo Telles 2,3 

Antimicrobial Activity of Cefoperazone-Sulbactam Tested against Gram-Negative Organisms from Europe, Asia-Pacific, and Latin America

Helio S. Sader

PlumX Metrics

DOI: https://doi.org/10.1016/j.ijid.2019.11.006

R: Dear reviewer, we included the references about study you proposed. See references N°66 and 67.

Thank you for your important suggestions.

Reviewer 3 Report

Comments and Suggestions for Authors

The present review is clear, objective, and introduces readers to the emerging subject of Acinetobacter baumannii resistance and the potential of mono and combination therapies against this prevalent bacterial pathogen. The reviewer recommends some minor adjustments before the manuscript is deemed suitable for publication.

General

1.      Please thoroughly review the manuscript to ensure that all microbial names are appropriately italicized.

2.      Change "in-vitro" to "in vitro" throughout the manuscript.

3.      All abbreviations must be fully explained initially. In this context, numerous abbreviations in the manuscript should be explained to ensure the clarity and comprehensibility of the manuscript.

4.      The reviewer recommends including a table or figure to visually depict all antimicrobials mentioned in the manuscript, along with their corresponding mechanisms of action and resistance mechanisms elucidated in Acinetobacter baumannii.

Abstract

1.      I propose commencing the abstract with a sentence addressing the global challenge of antimicrobial resistance, emphasizing its impact on mortality rates and economic costs. By highlighting the broad scope of this issue from a global perspective, we can immediately capture the attention of readers and underscore the urgency of addressing antimicrobial resistance in our research.

2.      Change the sentence “Cefiderocol, could play a crucial role in combination-regimen” to “Cefiderocol could play a pivotal role within combination therapy regimens.”

3.      Additionally, the abstract would benefit from a concluding sentence summarizing the key findings or implications of the study.

Introduction

1.      The reviewers strongly advise against abbreviating the name of Acinetobacter baumannii as 'AB,' as it does not adhere to international standards. Instead, please consistently use 'A. baumannii' throughout the manuscript text.

2.      In Table 1, the notation "1 g sulbactam component q6h" indicates the dosing regimen for the sulbactam component, which entails administering 1 gram of sulbactam every 6 hours. Is it correct?

2. Considerations and available data about new β-lactam agents

1.      The text would be enhanced by the inclusion of explanations regarding the classification of beta-lactamases.

2.      The information in the sentence 'The phenomenon of heteroresistance among CRAB isolates, defined as the presence of resistant subpopulations at a frequency of ≥1 in 10 to the power of 6,' needs to be corrected.

3.      The review requested that the author revise the following sentence: " The probability of target attainment (PTA) for 100% fT>MIC was >90% across all infection sites for pathogens with cefiderocol MICs of ≤4 μg/mL…," as it is confusing and requires clarification of the term "fT.

4.      The following sentence should be rewritten: “Two of them showed a similar rate of all-cause mortality (at days 28 and 30, respectively) between treatment groups [32,33], while in the others the cefiderocol-based regimen was an independent predictor of 30-day survival [10,11].”

3.    Place in therapy of traditional agents for treatment of CRAB

1.      Please split the following big sentence in two minor ones: Anyhow, in a prospective, observational, multicenter study, conducted on 180 patients with HAP due to MDR-AB, 44 patients were treated with a fosfomycin-containing regimen (29 in double and 15 in triple combination regimen) which turned to be a factor associated with 30-day survival (p < 0.001) [50]; moreover, in a case-series study conducted on 20 ICU patients with BSI due to pan-drug resistant (PDR)-AB, a fosfomycin-containing regimen (1 case in double regimen and 7 cases in regimens including at least three antibiotics) was associated with 28-day survival (p < 0.005) [51].

Conclusion

1.      The content presented in Table 3 would benefit from more detailed discussion within the main text, rather than being relegated to the conclusion section.

Comments on the Quality of English Language

The English should be checked to minimize errors and inconsistencies.

Author Response

The present review is clear, objective, and introduces readers to the emerging subject of Acinetobacter baumannii resistance and the potential of mono and combination therapies against this prevalent bacterial pathogen. The reviewer recommends some minor adjustments before the manuscript is deemed suitable for publication.

 R: Dear Reviewer, Thank you very much for the quick and thorough feedback. It'll really help us make the manuscript more accessible and engaging based on your important comments.

General

  1. Please thoroughly review the manuscript to ensure that all microbial names are appropriately italicized.
  2. Change "in-vitro" to "in vitro" throughout the manuscript.
  3. All abbreviations must be fully explained initially. In this context, numerous abbreviations in the manuscript should be explained to ensure the clarity and comprehensibility of the manuscript.
  4. The reviewer recommends including a table or figure to visually depict all antimicrobials mentioned in the manuscript, along with their corresponding mechanisms of action and resistance mechanisms elucidated in Acinetobacter baumannii. 

Abstract

  1. I propose commencing the abstract with a sentence addressing the global challenge of antimicrobial resistance, emphasizing its impact on mortality rates and economic costs. By highlighting the broad scope of this issue from a global perspective, we can immediately capture the attention of readers and underscore the urgency of addressing antimicrobial resistance in our research.
  2. Change the sentence “Cefiderocol, could play a crucial role in combination-regimen” to “Cefiderocol could play a pivotal role within combination therapy regimens.”
  3. Additionally, the abstract would benefit from a concluding sentence summarizing the key findings or implications of the study.

Introduction

  1. The reviewers strongly advise against abbreviating the name of Acinetobacter baumannii as 'AB,' as it does not adhere to international standards. Instead, please consistently use 'A. baumannii' throughout the manuscript text.
  2. In Table 1, the notation "1 g sulbactam component q6h" indicates the dosing regimen for the sulbactam component, which entails administering 1 gram of sulbactam every 6 hours. Is it correct?

  1. Considerations and available data about new β-lactam agents
  2. The text would be enhanced by the inclusion of explanations regarding the classification of beta-lactamases.
  3. The information in the sentence 'The phenomenon of heteroresistance among CRAB isolates, defined as the presence of resistant subpopulations at a frequency of ≥1 in 10 to the power of 6,' needs to be corrected.
  4. The review requested that the author revise the following sentence: " The probability of target attainment (PTA) for 100% fT>MIC was >90% across all infection sites for pathogens with cefiderocol MICs of ≤4 μg/mL…," as it is confusing and requires clarification of the term "fT.
  5. The following sentence should be rewritten: “Two of them showed a similar rate of all-cause mortality (at days 28 and 30, respectively) between treatment groups [32,33], while in the others the cefiderocol-based regimen was an independent predictor of 30-day survival [10,11].”

  1. Place in therapy of traditional agents for treatment of CRAB
  2. Please split the following big sentence in two minor ones: Anyhow, in a prospective, observational, multicenter study, conducted on 180 patients with HAP due to MDR-AB, 44 patients were treated with a fosfomycin-containing regimen (29 in double and 15 in triple combination regimen) which turned to be a factor associated with 30-day survival (p < 0.001) [50]; moreover, in a case-series study conducted on 20 ICU patients with BSI due to pan-drug resistant (PDR)-AB, a fosfomycin-containing regimen (1 case in double regimen and 7 cases in regimens including at least three antibiotics) was associated with 28-day survival (p < 0.005) [51].

Conclusion

  1. The content presented in Table 3 would benefit from more detailed discussion within the main text, rather than being relegated to the conclusion section.

R: here all the modifications on the manuscript

  • Lines 14-15, page 1: As suggested, we have included an impact statement on AMR mortality to stimulate reader interest in the impact of the issue.
  • Line 19, page 1: we modified the sentence as suggested.
  • Lines 27-28, page 1: We added concluding sentence in the abstract.
  • Introduction: “AB” abbreviation removed.
  • Table 1, page 2: Yes, the table means that the dosage of Sulbactam.
  • Lines 100-102, page 3: The general classification of beta-lactamases was included.
  • Lines 186-196, page 5: We have corrected the reported sentence so that it is clearer.
  • Lines 246-249, page 7: We appreciate your suggestion and we've reworked the following sentence to make it even better.
  • fT>MIC means the percentage of time that the free concentration remains above the minimum inhibitory concentration (%fT > MIC) of the pathogen.
  • Lines 391-398, page 10: As suggested, we have split the sentence into two different periods to make it more usable.
  • Table 3 lists all the antibiotics we've tested. We've added two more columns to include the mechanism of action and resistance in Acinetobacter baumannii.
  • Chapter 3, Place in therapy of traditional agents for treatment of CRAB: We've included the main characteristics of antibiotics in Table 3, not just in the conclusion.

Round 2

Reviewer 1 Report

Comments and Suggestions for Authors

Dear Authors,

The manuscript is well written, the additions are enough to have a more complete picture of this very interesting study.